# ALFA: Aligning LLMs to Ask Good Questions
# A Case Study in Clinical Reasoning

**Shuyue Stella Li**[1*]    **Jimin Mun**[2*]    **Faeze Brahman**[3]
**Pedram Hosseini**[4]    **Bryceton G. Thomas**[5]    **Jessica M. Sin**[5]    **Bing Ren**[5]
**Jonathan S. Ilgen**[1]    **Yulia Tsvetkov**[1]    **Maarten Sap**[2]
[1]University of Washington    [2]Carnegie Mellon University    [3]Allen Institute for AI
[4]Lavita AI    [5]Dartmouth Medicine
stelli@cs.washington.edu, jmun@andrew.cmu.edu
https://github.com/stellalisy/ALFA
https://huggingface.co/datasets/stellalisy/MediQ_AskDocs

## Abstract

Large language models (LLMs) often fail to ask effective questions under uncertainty, making them unreliable in domains where proactive information-gathering is essential for decision-making. We present **AL**ignment via **F**ine-grained **A**ttributes, (**ALFA**) a framework that improves LLM question-asking by (i) *decomposing* the notion of a "good" question into a set of theory-grounded attributes (e.g., clarity, relevance), (ii) controllably *synthesizing* attribute-specific question variations, and (iii) *aligning* models via preference-based optimization to explicitly learn to ask better questions along these fine-grained attributes. Focusing on clinical reasoning as a case study, we introduce the **MediQ-AskDocs** dataset, composed of 17k real-world clinical interactions augmented with 80k attribute-specific preference pairs of follow-up questions, as well as a *novel* expert-annotated interactive healthcare QA task to evaluate question-asking abilities. Models aligned with ALFA reduce diagnostic errors by 56.6% on *MediQ-AskDocs* compared to SoTA instruction-tuned LLMs, with a question-level win-rate of 64.4% and strong generalizability. Our findings suggest that explicitly guiding question-asking with structured, fine-grained attributes offers a scalable path to improve LLMs, especially in expert application domains.[1]

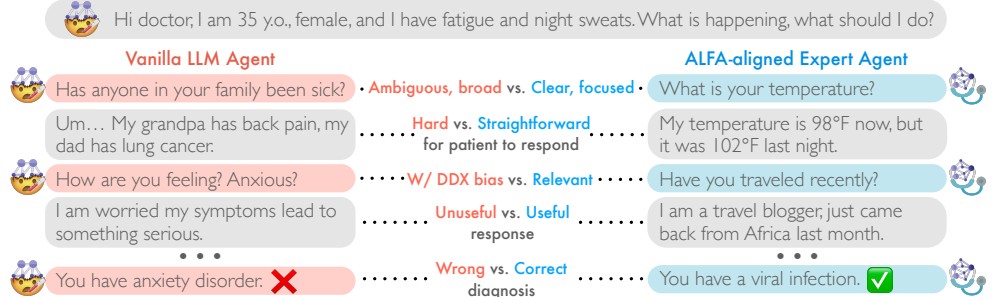

**Figure 1:** Effective information-seeking questions are crucial for clinical reasoning. ALFA-aligned models can ask better questions and lead to more accurate diagnosis.

## 1 Introduction

Interactive language models have demonstrated remarkable capabilities across numerous domains (OpenAI et al., 2024), yet *proactive* interaction abilities in high-stakes scenarios—clinical reasoning, legal analysis, investigative journalism—remains a challenge (Fung et al., 2024). A key obstacle is the ability of these models to recognize and anticipate missing or ambiguous information and proactively seek clarification (Li et al., 2024; Deng et al.,

---

[1]We release all data, code, and models for further research.

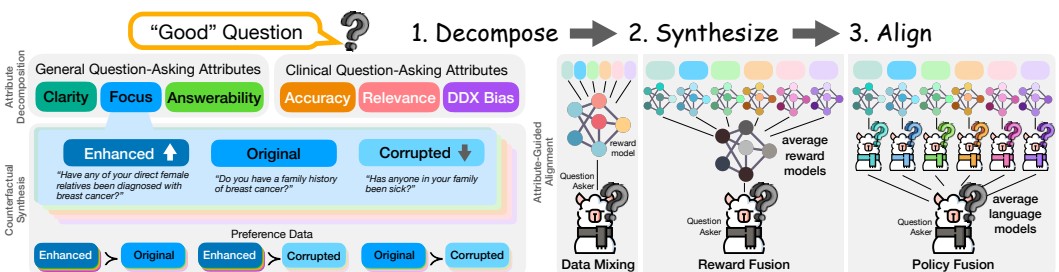

**Figure 2:** ALFA: decompose, synthesize, align.

2024). In clinical practice, for instance, physicians systematically ask patients questions to rule out or confirm relevant diagnoses (Richardson et al., 1995; Proffit, 2013). This iterative, information-seeking behavior is essential for accurate and safe decision-making. Similarly, for large language models (LLMs) to serve as *reliable* decision-support tools for clinicians, they must learn not only to provide answers, but also to identify when additional information is needed, and to ask follow-up questions that effectively explore and adjust possible hypothesis or reduce uncertainty (Figure 1).

However, there are two main challenges in building LLMs that ask good questions, especially in expert domains. First, defining a "good" question is inherently complex and context-dependent. In general, attributes such as *clarity*, *focus*, and *answerability* are essential (Heritage & Maynard, 2006; Roter & Hall, 1987; Freed, 1994; Searle, 1969); however, in domain-specific scenarios such as clinical reasoning, additional properties—*medical accuracy*, *diagnostic relevance*, and *mitigating differential diagnosis (DDX) biases*—are necessary for reducing diagnostic uncertainty (Richardson et al., 1995; Silverman et al., 2016; Heritage, 2010; Hall et al., 1995; West, 1984; Stivers & Majid, 2007; Ong et al., 1995). Second, instilling the ability to ask good questions in LLMs is technically non-trivial (Li et al., 2024; Johri et al., 2025; Zhang et al., 2024a). Naïve prompting strategies such as "*Ask a follow-up question if needed.*" may enhance interactivity but lack a principled foundation for defining a *good* question. We propose leveraging well-established general and task-specific principles from communication theory and psychology to improve LLMs' information-seeking abilities.

Methodologically, we introduce a general recipe to incorporate question-asking abilities into LLMs, focusing on clinical reasoning as a case study. Our method—**AL**ignment via **F**ine-grained **A**ttributes (ALFA)—relies on the idea that question quality can be improved by explicitly training models with data grounded in structured, theoretically motivated attributes. Our recipe proceeds in three steps:

1. *Decompose* the goal of asking "good" questions into structured, grounded attributes.
2. *Synthesize* counterfactual data by controllably altering any attribute (e.g. make *clearer*).
3. *Align* models using preference optimization algorithms to integrate the attributes and produce a final policy.

Since labeled conversational datasets containing follow-up questions along a variety of important attributes are scarce, ALFA exposes models to a much broader range of question-asking behaviors than what can be typically found in the wild, especially in specialized domains like clinical interactions.

We instantiate the above recipe with a focus on clinical reasoning, where question-asking is central to reducing diagnostic uncertainty and preventing errors. To this end, we construct a novel dataset, *MediQ-AskDocs*, containing 17k clinical interactions with follow-up questions from the r/AskDocs subreddit[2], paired with 80k synthesized counterfactual variants of these questions highlighting each attribute. These counterfactual pairs provide fine-grained training signals for preference learning (Li et al., 2025). Finally, we integrate the attribute-specific signals into a unified policy by combining all synthetic data into a single model, training separate reward models and merging them, or fusing attribute-specific policies. ALFA contrasts with coarse-grained preference learning, offering a more targeted way to refine question-asking behaviors.

---

[2]With medical experts verified following subreddit policy. See §7 for further discussion on setting.

As part of *MediQ-AskDocs*, we introduce a novel healthcare QA task of 302 expert-annotated clinical interaction scenarios to evaluate the proposed method. These scenarios are passed into MediQ (Li et al., 2024), an interactive clinical simulator, in which the models asks questions to the patient agent. ALFA-aligned models achieve a 64.4% win-rate in question-level evaluation and a 56.6% reduction in diagnostic errors, relative to baselines of SoTA instruction tuned LLMs. Beyond these empirical gains, our work presents a new paradigm for aligning language models to specialized domains by **decomposing** the complex goal of question-asking into attributes, **synthesizing** pairwise data in each attribute dimension, and **aligning** the model to jointly optimize the overall complex goal. This general approach to attribute-based question-asking alignment can be extended to many other domains where systematically eliciting information is key to reliable, effective decision-making.

## 2 Problem Statement

Aligning LLMs to ask good question requires reasoning along multiple attributes (e.g., accuracy, clarity, focus). However, most alignment paradigms treat these goals as monolithic, aggregating preferences into a single reward that conflates attributes and obscures their individual contributions. We formalize this challenge as follows:

Given a complex goal $G$ and a dataset with sparse labels $\mathcal{D}$, we aim to learn a policy $\pi$ that maximizes the composite reward $R(s, a)$, where:

- $s$: Current world state (e.g., information acquired so far, conversation history).
- $a$: Next action (e.g., follow-up question asked by the clinician agent $\pi$).

The key challenges lie in the complexity of the goal and the sparsity of labeled data. First, directly optimizing $R(s, a)$ is infeasible because human preferences for $R(s, a)$ are noisy and subjective. To address this, ALFA decomposes $G$ into $K$ attributes $\{A_1, \ldots, A_k\}$, each corresponding to a verifiable criterion with a reward function $R_k$. We constrain the selection of $A_k$ such that each $R_k(s, a)$ is more measurable compared to $R(s, a)$.

Second, we cannot observe *parallel* follow-up questions in natural conversations to construct preference pairs. To this end, we synthesize counterfactual synthetic data $\mathcal{D}^k_{synth}$ for each $A_k$, where

$$\mathcal{D}^k_{synth} = \{(a_i^{k+}, a_i^{k-}) | R_k(a_i^{k+}) > R_k(a_i^{k-})\}.$$

Lastly, ALFA uses a reward integration strategy $f$, where $R(s, a) = f(R_1(s, a), \ldots, R_K(s, a))$, to combine $\{R_1, \ldots, R_K\}$ into a policy $\pi$, such that:

$$\pi^* = \arg \max_\pi \mathbb{E}_{(s,a) \sim \pi}[f(R_1, \ldots, R_K)(s, a)],$$

optimizes performance on the complex goal $G$, aligning models to be better question-askers.

## 3 What Makes a "Good" Question?

To systematically improve LLM question asking, we define six key attributes grounded in cognitive science, psychology, and clinical communication research (Heritage & Maynard, 2006; Roter & Hall, 1987; Chouinard, 2007; Freed, 1994; Searle, 1969; Levinson, 2012). Unlike prior work that relies on implicit heuristics, we explicitly decompose question quality into interpretable, tangible attributes that enhance clinical reasoning.

**General Question-Quality Attributes.** Effective questions must be clear, targeted, and answerable to drive meaningful interactions. We select three core attributes:

1. *Clarity*, aiming to avoid ambiguity and unnecessary complexity (e.g., no jargons), ensuring precise communication (Heritage & Maynard, 2006; Roter & Hall, 1987; Burns et al., 2022);
2. *Focus*, directly addressing a specific information gap, yielding more informative responses (Ronfard et al., 2018; Gopnik & Wellman, 2012; Chouinard, 2007; Freed, 1994). E.g., "Has anyone in your family had breast cancer?" is superior to "Has anyone in your family been sick?"; and

3. *Answerability*, ensuring the question is both within the respondent's knowledge domain and appropriate for them to answer (e.g., asking about their symptoms and experiences rather than expecting them to provide medical diagnoses or knowledge that falls within the clinician's expertise) (Levinson, 2012; Keil et al., 2008; Searle, 1969).

**Domain-specific question-asking attributes.** In clinical reasoning, question-asking is a structured diagnostic skill. Drawing from clinical communication research (Richardson et al., 1995; Silverman et al., 2016; Heritage, 2010; Hall et al., 1995; West, 1984; Stivers & Majid, 2007; Ong et al., 1995; Proffit, 2013), we define three additional attributes essential for clinical decision-making:

4. *Medical Accuracy* requires alignment with established medical textbook knowledge & guidelines;
5. *Diagnostic Relevance* probes for symptoms, risk factors, or contextual details essential to refining differential diagnoses (DDX); and
6. *Avoiding DDX Bias* prevents suggestive or leading wording that could introduce cognitive biases and misguide diagnostic reasoning.

These six attributes form the foundation of ALFA, guiding question optimization to improve LLM reliability in interactive clinical reasoning.

## 4 ALFA Framework Overview

We now introduce ALFA, a structured recipe that decomposes the overall question-asking objective (§4.1), generates attribute-specific preference data (§4.2), and trains a policy that integrates the attributes (§4.3) to ask better follow-up questions.

### 4.1 Grounded Attribute Decomposition

We first decompose the concept of "good" clinical questions into the six attributes $A_k$ identified in §3 rather than relying on implicit heuristics or coarse scoring. This decomposition enables two advantages: (1) *attribute-specific training signals* that isolate distinct aspects of question quality, and (2) a *controlled preference structure* for fine-grained alignment, guiding the next stages of data generation and model alignment.

### 4.2 Attribute-Specific Data Generation

Real-world clinical datasets are scarce, private, and rarely contain annotations distinguishing, for instance, *clear* vs. *ambiguous* questions (Mireshghallah et al., 2023; Ramesh et al., 2024). Therefore, we generate *synthetic preference data* by (1) collecting authentic clinical posts (see §5.3 for dataset curation details) and (2) using an LLM to generate counterfactual variants along each attribute.

**Counterfactual Perturbation.** For each question $a_i$ in the dataset, we prompt an LLM to create "enhanced" and "corrupted" variants $(a_i^{k+}, a_i^{k-})$ that explicitly alter only one attribute $k$ at a time (e.g., rewriting the original question to be more clear/ambiguous) while keeping others consistent. This enables us to create controlled preference pairs, where one version of a question $a_i^{k+}$ is better aligned with a specific attribute than the other, $a_i^{k-}$.

**Verification & Filtering.** We use an **LLM-judge** (Zheng et al., 2023) to verify that the generated perturbations reflect their intended modification. With original question $a_i$, we obtain an enhanced question $a_i^{k+}$ and a corrupted question $a_i^{k-}$, resulting in three preference pairs: $(a_i^{k+}, a_i)$, $(a_i^{k+}, a_i^{k-})$, and $(a_i, a_i^{k-})$. Given each pair, we provide additional context[3] to the LLM-judge, and ask the judge to compare the pairs in the specified attribute dimension (e.g., which question is *clearer*). If the judge's decision matches the intended perturbation direction—verifying that $R_k(a_i^{k+}) > R_k(a_i^{k-})$—we retain the sample; otherwise, we discard it. This filtering step removes inconsistencies and ensures that our synthetic preference data provides reliable supervision for model alignment. See details in Appendix C.1.

---

[3] Parsed conversation conclusions from future turns.

### 4.3 Attribute Integration Strategies

We now *combine* signals from the attribute-specific data so that the model produces questions that optimize for all attributes. Standard preference optimization algorithms DPO (Rafailov et al., 2024) and PPO (Schulman et al., 2017) allow distinct *Points of Integration* (POI) as described below[4]:

(1) **Data Mixing** pools all data into one training set and uses standard DPO/PPO. This treats each attribute-specific comparison as part of a larger set of "better vs. worse" pairs, enabling a single policy to learn from all attributes at once (Wang et al., 2023b; Lambert et al., 2024).

(2) **Reward Fusion** trains separate reward models (RM), then averages the RM *weights* in an overall RM which can then be used in PPO to align a final policy (Ramé et al., 2024).

(3) **Policy Fusion** trains separate policies or preference models, each specialized for one attribute, and then combine the model weights by averaging or taking a linear combination (Jang et al., 2023). This strategy offers the highest degree of parallelism and maximally preserves attribute strengths and interpretability.

Comparing these integration strategies informs how best to reconcile multiple, sometimes competing, objectives (e.g., *focus* vs. *avoiding DDX bias*) and thereby produce consistently high-quality diagnostic questioning.

## 5 Experimental Setup

### 5.1 *MediQ-AskDocs* Dataset Curation

We obtain data from `r/AskDocs`, a public online health forum, and filter for conversation threads where (i) the patient posts a health inquiry and engage with another user to discuss the issue, and (ii) another user *asks follow-up questions* to acquire more information. The resulting dataset contains 13,496 unique posts, 17,425 threads, and 24,263 questions. For creating the counterfactual perturbations (§4.2), we sample 4463/433/620 questions for the train/dev/test splits. More details on dataset curation and sampling are in Appendix B.

### 5.2 Experiments

With the goal of instilling question-asking ability in LLMs in the clinical reasoning domain, we structure our experiments to progressively address two key questions: *(1) Does the entire* ALFA *pipeline improve clinical question-asking performance?* and *(2) To what extent is every component of* ALFA *necessary for improving performance?* Accordingly, we organize our evaluation as follows:

**1. Overall Performance.** We first confirm that ALFA meaningfully reduces diagnostic errors and elicits better questions in an interactive clinical scenario (§6.1).

**2. Key Pipeline Components.** We isolate the two core components of ALFA:

- We compare the *decomposed* attributes (e.g., clarity, answerability) to a "coarse" attribute (simply "good or bad") to examine the effect of theory-grounded decomposition (§6.2).
- We compare preference tuning to supervised fine-tuning (SFT) on *the same* synthetic data, revealing how pairwise reward signals refine question-asking beyond what SFT alone can achieve (§6.3).

**3. Ablation Studies.** We dissect each design choice to identify its role in the observed improvements. We compare *attribute-integration strategies*, apply *quality filter* on the synthetic data, examine *data perturbation directions* (corruption vs. enhancement), ablate *individual attributes*, and test *out-of-distribution generalization* on a separate diagnostic task[5] (§6.4).

---

[4]While reward sum, which trains attribute-specific reward models then combines the reward scores (e.g., via a learned linear combination or average) to produce a final scalar reward (Wu et al., 2023; Wang et al., 2024c), remains an option, we do not utilize this strategy due to high compute cost of loading all reward models during PPO without substantial performance improvements (Rame et al., 2024; Shi et al., 2024).

[5]All ablations are done with DPO due to lighter compute requirements unless otherwise specified.

Collectively, these analyses elucidate how components in ALFA—attribute decomposition, data synthesis, preference tuning—work together to improve question-asking in high-stakes clinical contexts. See implementation details in Appendix C.

## 5.3 Evaluation

We evaluate the aligned models along two fronts: **Direct Question Quality**, measured through expert human annotations and automatic LLM-judge comparison for the overall question quality, and **Clinical Decision Impact**, quantified by how well the model questions help reduce diagnostic errors.

**Direct win-rate with LLM-judge.** We create an LLM-judge to compare pairs of questions, adopting prompt structures from Li et al. (2023) and Dubois et al. (2023). Specifically, we measure the percentage of times our aligned models' questions are preferred over those of the baseline instruction-tuned models (llama-3.2-3b-Instruct and llama-3.1-8b-Instruct) by the judge and report win-rate. Each comparison is carried out by `gpt-4o` with rationales, permuting the presentation order in each pair to mitigate ordering bias. As an validity check, the LLM-judge assigns a higher win-rate to questions from verified experts than those from non-experts (Appendix D), suggesting our evaluation aligns with domain expertise.

**Expert manual evaluation.** To assess the quality of generated questions and further validate the LLM-judge, we conduct manual preference rankings with *three medical experts* from our research team and compute win-rate of select models based on majority vote. See Appendix F for further details.

**Interactive diagnostic accuracy.** We use the MediQ interactive framework (Li et al., 2024)— patient-clinician simulator—to holistically evaluate ALFA in a more realistic setting. MediQ presents some initial information $x^0$ (often the patient's chief complaint), a medical inquiry $\kappa$, and tests an expert agent's ability to ask follow-up questions $a$ to the patient until it has enough information to make a diagnosis $y$. We replace the question generator module in the MediQ expert agent with models trained with ALFA, while keeping all other modules (patient system and diagnosis generator) consistent. We quantify the utility of the question generator with the accuracy of the diagnosis $y$.

**The *MediQ-AskDocs* task.** MediQ is compatible with any QA task with contextual information. We introduce **a novel healthcare QA task** as part of the *MediQ-AskDocs* dataset: 302 consumer healthcare multiple choice questions manually annotated by medical experts. The task is automatically generated by o1 using the test split of *MediQ-AskDocs* (§5.1) and achieves 85.9% agreement with majority voted manual annotations from medical experts. See Appendix E for task construction and annotation details. Additionally, we use MedQA (Jin et al., 2020) with MediQ to examine the models' ability to generalize out-of-domain.

# 6 Results & Analysis

## 6.1 ALFA Improves Overall Question Asking

We begin by comparing ALFA with two baselines: (1) the base instruction-tuned models (llama-3.2-3B-Instruct and llama-3.1-8B-Instruct) and (2) models trained via supervised fine-tuning (SFT) on the human-written questions. The Policy Fusion attribute integration strategy (§4.3) is reported for both ALFA-DPO and ALFA-PPO in Table 1.

Our results confirm that fine-tuning the human-written questions in *MediQ-AskDocs* (SFT) already outperforms base models, establishing usefulness of the dataset. Notably, superior performance of the ALFA-aligned models shows *explicitly modeling structured, theory-grounded attributes substantially boosts both question quality and diagnostic accuracy*, reducing diagnostic errors by 56.62% (increasing accuracy by 21.5%).

| Model | Size | Win-rate | MediQ-AD |
|---|---|---|---|
| Base Model | 3B | 50.00 | 73.51 |
| | 8B | 50.00 | 72.52 |
| SFT | 3B | 61.04 | 78.98 |
| | 8B | 58.23 | 85.08 |
| ALFA-DPO | 3B | **64.97** | **87.75** |
| | 8B | **65.13** | **88.08** |
| ALFA-PPO | 3B | 64.84 | 86.75 |

**Table 1:** Main results. ALFA models consistently outperform base instruct and SFT models. Note that win-rates for base models are set to 50% to represent equal preference when comparing against themselves.

| Model | Size | Win-rate | MediQ-AD |
|---|---|---|---|
| Coarse | 3B | **65.89** | 83.77 |
| | 8B | **66.05** | 85.76 |
| ALFA (Fine-Grained) | 3B | 64.97 | **87.75** |
| | 8B | 65.13 | **88.08** |

**Table 2:** Fine-grained (ALFA) vs. Coarse Attributes. Fine-grained attributes lead to better downstream diagnostic accuracy and similar win-rates compared to coarse-grained objective. Expert evaluation for 3B coarse vs. fine-grained shows identical win-rate against the base model: **59.4%**.

## 6.2 Fine-Grained Attributes Outperform Coarse Attribute

The core idea of ALFA is to decompose a complex goal into structured, theory-grounded attributes rather than treating it as one coarse objective. We examine the role of attribute decomposition by comparing alignment with **fine-grained** attributes (ALFA) with a simpler approach that optimizes models on a **coarse** "better" vs. "worse" distinction (Geng et al., 2025). Both models undergo the same counterfactual data generation and alignment processes, but the coarse model lacks explicit attribute separation.

Results in Table 2 show that ALFA, trained with fine-grained attributes, achieves higher performance compared to models trained on coarse attributes on MediQ diagnostic accuracy, while the two methods show comparable LLM-judge win-rates. Aligning models based on structured attributes guides the reasoning process in a theory-grounded way, resulting in substantially better downstream performance. Additionally, qualitative analysis (Figure 3) shows that the coarse-aligned model tends to ask more superficial questions such as waiting for 8 weeks and check back, while the ALFA-model's question, "*Was this the only issue that was found?*", indicates some form of reasoning to rule out other factors that might have contributed to the patient's concern. Further, coarsely aligned models generalize poorly to out-of-distribution tasks (§6.4).

Patient Context:
What is the treatment for sever hypovitaminosis D? My 25(OH)D test result was 4.68 ng\/ml. My doctor prescribed four IM injections of 300,000 IU vitamin D within four weeks, and use of oral supplements after that. isn't four injections of 300,000 IU in this short time too much? do I really need this much vitamin D? should I get a second opinion?

Coarse-Aligned LM Output:
Are you getting it checked again in 8 weeks?

ALFA-Aligned LM Output:
Was this the only issue that was found?

**Figure 3:** Qualitative example contrasting models aligned with fine-grained and coarse attributes. ALFA model tends to ask more logical questions.

## 6.3 Preference Tuning Outperforms SFT

Another core advantage of ALFA is its use of pairwise preference learning, which allows models to refine question-asking beyond what supervised fine-tuning (SFT) achieves. To test whether ALFA's improvements come solely from exposure to diverse synthetic data or from learning structured preferences, we compare ALFA with models fine-tuned on real data (SFT-Real), synthetic enhanced data (SFT-Synthetic), and both (SFT-Combined).

As shown in Table 3, ALFA outperforms SFT-Combined despite learning from the same data, confirming that learning **directional differences** from pairwise comparisons is

| Model | Size | Win-rate | MediQ-AD |
|---|---|---|---|
| SFT-Real | 3B | 61.04 | 78.98 |
| | 8B | 58.23 | 85.08 |
| SFT-Synthetic | 3B | 62.50 | 82.12 |
| | 8B | 60.32 | 85.76 |
| SFT-Combined | 3B | 62.50 | 83.77 |
| | 8B | 54.68 | 85.76 |
| ALFA | 3B | **64.97** | **87.75** |
| | 8B | **65.13** | **88.08** |

**Table 3:** Preference tuning is crucial in improving model performance. Supervised fine-tuning on the same synthetic data does not show as much performance gain.

key to better question-asking. These results highlight pairwise contrastive optimization as a necessary step for models to learn how to ask better follow-up questions.

| *PO | POI | Size | Win-rate | MediQ-AD | Win-rate (human) |
|-----|-----|------|----------|----------|------------------|
| DPO | Data | 3B | 68.55 | 85.01 | 52.00 |
| | | 8B | **68.23** | **88.74** | — |
| | Policy | 3B | 64.97 | **87.75** | 41.00 |
| | | 8B | 65.13 | 88.08 | — |
| PPO | Data | 3B | 97.34 | 84.77 | **75.00** |
| | Reward | 3B | **97.98** | 84.44 | 74.00 |
| | Policy | 3B | 64.84 | 86.75 | 40.00 |

**Table 4:** Attribute integration strategies.

| Model Variant | Model Size | Win Rate | MediQ-AD Acc. (%) |
|---------------|------------|----------|-------------------|
| ALFA | 3B | 64.19 | 85.43 |
| -Unfiltered | 8B | 63.31 | 86.09 |
| ALFA | 3B | **64.97** | **87.75** |
| | 8B | **65.13** | **88.08** |

**Table 5:** Synthetic data quality. Filtering slightly improves diagnostic accuracy.

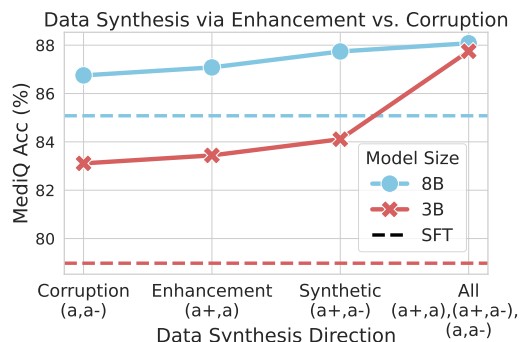

**Figure 4:** All synthetic data directions are helpful. Including corruptions, enhancements, and the original data shows the best performance.

## 6.4 Ablation Studies

**I. When to integrate the attributes?** We now examine the effect of various attribute integration strategies from §4.3: **data**-mixing, **reward**-fusion (PPO only), and **policy**-fusion. Table 4 reveals three key findings. First, data-mixing achieves higher question win-rate but yields lower diagnostic accuracy than policy-fusion, and reward-fusion in PPO mirrors data-mixing patterns. This suggests that *greater attribute separation leads to improved performance*, consistent with prior observations in Yang et al. (2025). Second, The LLM-judge scores had substantial agreement with human expert assessments for strategies with higher win-rate (Gwet's AC1 score of .68 and .72 each for ALFA-PPO-Reward and ALFA-PPO-Data), establishing the LLM-judge as a reliable proxy for human assessment in this simulated interaction environment. The high LLM-judge score of ALFA-PPO-Reward and ALFA-PPO-Data are echoed in the human evaluation with the highest win-rates of 74% and 75% respectively. See Appendix F and G for further details on annotations and qualitative analysis.

**II. Gains from Synthetic Data Quality.** ALFA relies on synthetic question perturbations to expose models to counterfactual scenarios. To ensure data quality, we filtered out 13.9% of generated pairs where LLM-judge ratings misalign with intended perturbation directions (§4.2). Filtering slightly improves both question quality and diagnostic accuracy, emphasizing the value of high-quality synthetic data (Table 5).

**III. Synthetic Corruption vs. Enhancement.** In the counterfactual pairwise data generation stage of ALFA, each original sample $a$ is synthesized in two directions along each attribute dimension (e.g. *more relevant* and *less relevant*) to get $a^+$ and $a^-$. In this section, we examine how the generation direction—*enhanced* ("more X") vs. *corrupted* ("less X")—influence performance. Specifically, we compare models trained with **corruption** only pairs $(a, a^-)$, **enhancement** only pairs $(a^+, a)$, **synthetic** corruption and enhancement pairs $(a^+, a^-)$, and **all** of the above. In Figure 4, we find while all directions are beneficial, combining all three pairs brings the most gains, especially apparent in the smaller 3B model.

| Attribute | Win-rate | MediQ-AD |
|---|---|---|
| No Accuracy | 63.95 | 84.11 |
| No Answerability | 64.60 | 84.11 |
| No Avoid DDX Bias | 62.58 | 81.79 |
| No Clarity | 62.74 | 85.10 |
| No Focus | 63.95 | 84.44 |
| No Relevance | 63.39 | 84.44 |
| General | 62.90 | 81.79 |
| Clinical | 66.05 | 86.09 |
| All | **64.97** | **87.75** |

**Table 6:** Policy Fusion DPO w/ attribute groups.

**ALFA with General Attributes (ALFA-General):**
```
Did they treat you for mono?
Is she in school?
```
**ALFA with Clinical Attributes (ALFA-Clinical):**
```
Did the pain worsen or improve with NSAIDs?
What do you think about the diagnosis of febrile
convulsion?
What is your age, sex, medical history, and
medications?
```

**Figure 5:** Models aligned with general vs. clinical attributes show distinct behaviors. ALFA-General: clear and focused, but less relevant and contains DDX bias ("*mono*"); ALFA-Clinical: professional but uses medical terms hindering answerability.

**IV. Attribute-Specific Influences.** A key component of ALFA is the explicit decomposition of question-asking into fine-grained, theory-grounded attributes. To assess their individual contributions, we conduct an ablation study where we remove one attribute at a time and evaluate the model's performance. We also compare general question-asking attributes (clarity, focus, answerability) with clinical attributes (medical accuracy, diagnostic relevance, avoiding DDX bias).

Table 6 shows that removing any attribute leads to performance drops, confirming their importance in clinical question-asking. Clinical attributes have a stronger impact on MediQ accuracy. Avoiding DDX bias—factors such as premature closure and availability bias leading to incomplete/incorrect differential diagnoses—is the most critical, suggesting that models need explicit training to counteract cognitive biases. Qualitatively, the questions generated by models aligned with general attributes vs. clinical domain-specific attributes show distinct styles (Figure 5), highlighting the impact of feature selection. These results validate ALFA's structured attribute alignment, demonstrating that both domain-specific capabilities and general question quality contribute to effective clinical decision-making.

**V. ALFA Models Robustly Generalize to Out-of-Distribution Settings.** We further assess ALFA's ability to generalize beyond the *MediQ-AskDocs* task by evaluating on a more challenging clinical reasoning benchmark *unseen* during training, MedQA (Jin et al., 2021), in the same MediQ-style interactive setting. Across intergration strategies, ALFA-aligned models generally outperform or match the coarse-alignment baseline when moving to MedQA (Figure 6). These suggests that learning from structured, theory-grounded attributes can enhance an LLM's robustness in new and more diverse clinical settings, highlighting ALFA's potential for broader applicability in real-world clinical scenarios.

**VI. ALFA Outperforms State-of-the-art General & Medical LLMs** Lastly, we compare ALFA-aligned models to both closed-source general purpose models and medical-specific models by plugging these models into MediQ as the question-generator. Table 7 shows that ALFA shows superior downstream task utility, substantially outperforming even much larger SOTA (general and medical) LLMs.

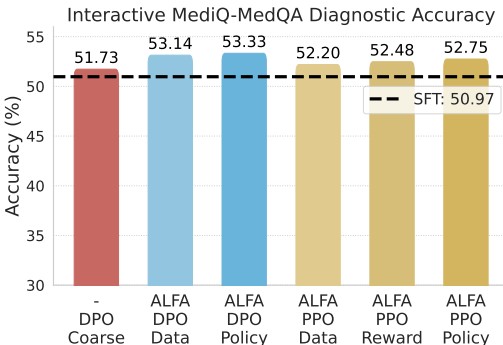

**Figure 6:** 3B Model performance on the interactive MediQ-MedQA task. Models aligned with ALFA are more robust to out-of-distribution data.

| Model | Diagnostic Acc. (%) |
|---|---|
| ALFA-DPO-8B | **88.1** |
| GPT-4o | 79.8 |
| Gemini-2 | 79.8 |
| o3-mini | 71.2 |
| Alpacare-7B | 74.8 |
| Alpacare-13B | 73.5 |
| MedAlpaca-13B | 68.5 |
| ClinicalCamel-70B | 70.5 |
| Meditron-7B | 70.2 |
| Meditron-70B | 72.5 |
| PMC-LLaMA-7B | 71.9 |
| PMC-LLaMA-13B | 69.9 |

**Table 7:** Diagnostic Accuracy of Various Models

## 7 Related Work

LLMs have the potential to significantly transform medicine by enhancing personalized care and accessibility (Shanmugam et al., 2024). Models trained with medical data contain rich medical knowledge (Singhal et al., 2025; Lewis et al., 2020; Chen et al., 2023; Labrak et al., 2024; Singhal et al., 2023; Brin et al., 2023); however, systematic evaluations reveal persistent weaknesses in instruction-following, multi-hop reasoning, and the nuanced pragmatics that arise in real clinical encounters (Hager et al., 2024; Arroyo et al., 2024; Nov et al., 2023; Zhang et al., 2014). These shortcomings are partly masked by the dominance of static, single-turn medical QA benchmarks, on which current models already achieve near-saturated scores (Jin et al., 2020; Pal et al., 2022). Recent work has moved away from the static single-turn paradigm and highlight *proactive information-seeking* as a prerequisite to reliable and effective clinical reasoning (Li et al., 2024; Hu et al., 2024b). CoAD (Wang et al., 2023a) is an early exploration toward interactivity, yet it operates in a markedly simplified symbolic environment: the agent selects from a small closed set of symptoms and diseases under dense supervision—impossible to obtain in real-life interactions. Crucially, the setting eliminates the linguistic realization entirely as the "question" is an index to a symptom list. ALFA tackles the more demanding challenge in the open-text regime: teaching an LLM to decide simultaneously what information remains most diagnostically valuable and how to ask for it in a way that is bias-sensitive and pragmatically appropriate.

Methodologically, our approach builds on recent data-centric alignment techniques that create synthetic preference signals for alignment (Li et al., 2025; Mishra et al., 2024a; Ding et al., 2024; Park et al., 2024). Inspired by prior work on multi-objective RLHF (Zhou et al., 2023b; Wu et al., 2023), we extend PPO (Ouyang et al., 2022; Christiano et al., 2017) and DPO (Rafailov et al., 2023) to align models with attribute-specific datasets, and uniquely compare different integration points of the fine-grained preference signals (Rame et al., 2024; Chronopoulou et al., 2023; Wang et al., 2024a).

## 8 Discussion

Effective question-asking is a fundamental yet underdeveloped capability in large language models, particularly in high-stakes domains like clinical reasoning. We proposed ALFA, a framework that explicitly teaches models to ask better questions by decomposing question quality into theory-grounded, fine-grained attributes and aligning them through preference-based optimization, rather than treating such nuanced and complex goal as a monolithic objective. We introduced *MediQ-AskDocs*, a comprehensive dataset of training data, preference data, and a healthcare QA task, showing that models trained with ALFA substantially outperform baselines. While focused on medicine as a case study, ALFA is a general recipe adaptable to any field where clear, targeted questioning is essential, paving the way for interactive and reliable systems.

**Future Directions.** While ALFA demonstrates significant improvements in clinical question-asking within controlled scenarios, several important directions warrant exploration. First, incorporating contextual factors that shape real clinical reasoning—such as care setting constraints (rural vs. urban hospitals), available diagnostic resources, and physician-patient relationship history—could enhance the framework's real-world applicability. Future work could explore dynamic weighting mechanisms conditioned on these contexts to improve attribute integration. Additionally, integrating multimodal inputs beyond text, such as tone analysis, non-verbal cues, and existing electronic health record data, could better mirror human clinical interactions. The ALFA framework could also benefit from incorporating collaborative decision-making elements, where models learn to ask patients about their own hypotheses or concerns, and to leverage input from healthcare team members. Finally, extending ALFA to other high-stakes domains requiring systematic information-gathering—such as legal discovery, investigative journalism, or financial risk assessment—could validate its generalizability beyond healthcare while revealing domain-specific attribute requirements.

## Limitations

**Manual attribute selection.** ALFA requires manual selection of attributes when adapting to new expert domains. While it offers a structured framework, determining which attributes are essential still depends on domain expertise. However, ALFA can also help evaluate attribute necessity across different fields.

**LLM dependence for counterfactual generation.** The counterfactual perturbation step relies on LLMs to generate and evaluate counterfactual question variants (specifically, `meta-llama/Llama-3.1-405B-Instruct-FP8`), assuming they correctly interpret attributes like clarity and relevance. Future work should incorporate human verification of counterfactuals and attribute-level rankings.

**Subjectivity in human annotation.** Evaluating follow-up questions is inherently subjective and scenario-dependent. Some annotators expressed difficulty in ranking questions, stating that "none of the questions were good" or that "all of them were acceptable." To ensure annotation quality, we implemented a four-question screening test with known ground-truth answers, filtering out annotators who failed to meet a predefined accuracy threshold. However, this approach does not fully eliminate the risk of variability in domain expertise.

**Data and scope.** Our dataset is derived from online health forum discussions (`r/AskDocs`) rather than in-person clinician-patient dialogues in a hospital setting. While this source provides diverse real-world medical inquiries, it does not fully capture the structured questioning strategies used in professional clinical settings. Thus, while ALFA offers a strong technical foundation for studying medical question-asking, it should not be viewed as a direct replacement for physician training or real clinical interactions. Expanding to EHR-based or in-hospital dialogue datasets would improve clinical applicability.

The other presumption underlying this work is that all of these medical problems have verifiable solutions. In actual medical practice—particularly in settings where problems are acute and undefined—it is quite likely that clinicians will come to slightly or entirely different solutions for the same problem (based on their own experiences or expertise).

## Ethics Statement

ALFA aims to improve LLM-driven question-asking in clinical reasoning, but its development and potential deployment pose potential ethical risks related to misinformation, bias, privacy, and regulatory compliance.

A primary concern is misinformation and overreliance on AI-generated questions. While ALFA improves question quality, it does not provide any sense of guarantee on factuality. If used without human oversight, it could generate misleading, irrelevant, or overly confident questions, potentially influencing clinical decision-making and leading to misdiagnosis or unnecessary medical interventions.

Bias in training data and evaluation is a key risk. ALFA, trained on `r/AskDocs`, may not represent diverse populations, conditions, or expert strategies, leading to systematic biases that reinforce healthcare disparities. U.S.-based, English-speaking annotators further limit generalizability. Reliance on LLM-judges may introduce automation bias, reinforcing subtle inaccuracies. Future work should expand datasets and evaluation to more diverse populations and implement bias mitigation strategies.

Privacy and data security are additional risks. Although ALFA does not process private medical records, future adaptations using clinical data or electronic health records (EHRs) must protect sensitive patient information. Transparent data governance frameworks and strict access controls are necessary for responsible use in healthcare applications.

ALFA is intended as a technical contribution to the field of computer science rather than a standalone clinical tool. The framework must be integrated with human-in-the-loop supervision, where clinicians retain final decision-making authority. Future work should explore uncertainty calibration, ethical safeguards, and regulatory alignment to ensure safe, fair, and reliable AI-assisted clinical reasoning.

## Acknowledgements

This research was developed with funding from the Defense Advanced Research Projects Agency's (DARPA) SciFy program (Agreement No. HR00112520300). The views expressed are those of the author and do not reflect the official policy or position of the Department of Defense or the U.S. Government. This material is based upon work supported by the Defense Advanced Research Projects Agency and the Air Force Research Laboratory, contract number(s): FA8650-23-C-7316. Any opinions, findings and conclusions, or recommendations expressed in this material are those of the author(s) and do not necessarily reflect the views of AFRL or DARPA. We would like to thank Dr. Saeed Hassanpour and the Dartmouth Center for Precision Health and Artificial Intelligence (CPHAI) for helping to facilitate our collaboration with the medical professionals team in partnership with Lavita AI.

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

## A  Extended Related Works

**Clinical LLMs.**  LLMs have potential to highly impact medicine (Moor et al., 2023; Thirunavukarasu et al., 2023) from personalizing care to improving accessibility (Rodriguez et al., 2024; August et al., 2023). Thus, many language models have focused on clinical knowledge and usage including closed-sourced Med-PaLM 2 (Singhal et al., 2025) to open models such as BioGPT (Lewis et al., 2020), Meditron (Chen et al., 2023), and BioMistral (Labrak et al., 2024) to name a few (Toma et al., 2023; Zhou et al., 2023a). More recently, many non-medical, general purpose models such as OpenAI's o1 have outperformed medically adapted models (Xie et al., 2024; Jeong et al., 2024). These models have shown human-level performance on MedQA (Jin et al., 2020) and other medical knowledge benchmarks (Singhal et al., 2023) and some have even shown to provide human-level soft skill such as empathy (Brin et al., 2023).

**Clinical Reasoning and Question-asking of LLMs.**  Reasoning abilities of these systems, especially under complex, high-stakes demands of medical interaction fulfilling various intentions of users (Nov et al., 2023; Zhang et al., 2014) require further attention. More specifically, clinical reasoning requires ability to ask effective questions (Silverman et al., 2016), crucial for information gathering phase with iterative hypotheses testing and updating. Shaikh et al. (2024) highlighted general lack of question-asking by LLMs in various contexts. While prior works have focused on improving question-asking of LLMs (Andukuri et al., 2024; Rao & Daumé III, 2018) with some focused diagnostic conversations, these works have been limited to rule-based, toy scenarios (Hu et al., 2024b), or prompting-based techniques (Li et al., 2024). Further, ALBA (Varadarajan et al., 2024) examined question-asking under mental health assessment setting. Thus, our work expand on such prior works towards a more flexible medical dialogue system with effective question-asking under various real-world user queries.

**Alignment Methods.**  Methodologically, our work adopts various alignment algorithms, especially PPO (Ouyang et al., 2022; Christiano et al., 2017) and DPO (Rafailov et al., 2023). Moreover, to integrate complex and nuanced preferences, multi-objective settings have been explored including MODPO (Zhou et al., 2023b) and MORLHF (Wu et al., 2023; Bai et al., 2022). Additionally various works have highlighted efficient methods to integrate multiple objectives, for example, through combining reward model or adapter weights (Rame et al., 2024; Chronopoulou et al., 2023; Ramé et al., 2024).

**Synthetic Data.**  With the growing capabilities of LLMs and to supplement human data, which can be sparse, synthetic data generation with LLMs have become a popular method (Long et al., 2024; Xu et al., 2024), especially to perform task-specific post-training (Kim et al., 2024). Synthetic data is especially appealing in healthcare domain as privacy issues can make data access prohibitive (Ramesh et al., 2024; Xin et al.; Murtaza et al., 2023). Thus, synthetic data generation through both rule-based methods (Fansi Tchango et al., 2022) and LLMs (Oh et al., 2024; Mishra et al., 2024b; Yao et al., 2024; Wang et al., 2024b) have been explored for various medical tasks. However, data to investigate question quality, especially in medicine, remains under-explored and our data generation method addresses this gap.

**Evaluation Frameworks.**  To assess LLMs in various medical tasks, many different evaluation frameworks have been developed, typically consisting of static, single-turn question-answering task based on multiple choice questions (Jin et al., 2020; Pal et al., 2022; Jin et al., 2019; Rawat et al., 2024). However, with the advancement of LLMs, there has been a growing need to evaluate LLM agents beyond simple demonstration of knowledge (Thirunavukarasu et al., 2023). MediQ (Li et al., 2024) proposes to evaluate LLMs' information-seeking ability through interactive clinical reasoning tasks, and constructs a benchmark based on MedQA

(Jin et al., 2020) leveraging information asymmetry at benchmark construction time and at inference time. MEDIC (Kanithi et al., 2024) explores comprehensive assessment of LLMs using methods such as LLM-as-a-judge (Zheng et al., 2023). Concurrent work HealthQ (Wang et al., 2024d) analyzes attribute related factors in their evaluation, but lacks theory-grounding and does not propose methods to specifically improve the attributes. Our work builds on prior works to comprehensively assess LLM's ability to seek information towards effective clinical communication utilizing LLM-as-a-judge and adopting MediQ framework with a newly generated set of task-specific multiple choice questions (Yao et al., 2025).

## B   Dataset Curation

*HealthQ* **Dataset.**   To study clinical conversations, we utilize data from publicly available online health forum `r/AskDocs`[6], a subreddit consisting of both lay-users and expert users[7]. We parsed each subsequent comments as a single thread and consider such threads as conversations between users. Since we are interested in clinical followup questions, we first selected threads where the first followup comment from the community contained sentences ending with question marks and further decomposed each conversation into atomic questions, conclusions, and presence of positive feedback from the post author (e.g., thank you) using GPT-4o[8]. The resulting dataset contained 17,425 threads, 13,496 unique posts, and 24,263 questions.

**Synthetic Data.**   We sampled questions from the above conversations to build a seed set for synthetic data generation. To ensure balanced quality for both corruption and enhancement in contrastive learning, we used proxy measures such as the expert verification status of the question author, the outcome of the conversation (e.g., final conclusions), and positive feedback from post authors. This resulted in 8 proxy quality groups. We evenly sampled from these groups, with the test set including only questions from threads with final conclusions. We created distinct train, validation, and test sets containing 4,463; 433; and 620 questions, respectively, ensuring no post overlap.

**Data Quality.**   We employed three key strategies to ensure data quality: (1) r/AskDocs has strict anti-misinformation policies and expert verification processes, (2) previous studies have validated the medical quality of responses in this subreddit, and (3) we filtered for samples with positive feedback and clear conclusions to ensure conversation quality.

## C   Implementation Details

### C.1   Counterfactual Perturbations

To generate counterfactual perturbations, we use `meta-llama/Llama-3.1-405B-Instruct-FP8` with VLLM on 8 A100 80GB GPUs with a temperature of 1.0 and max generation length of 512. See Appendix H.1 for an example prompt.

**Counterfactual Verification & Filtering**   To verify the quality of generated counterfactual perturbations, we use LLM-judge to rank the generated questions in the perturbed attribute (e.g., accuracy). Furthermore, we used LLM-judge result to filter generated data in § 6.4. To avoid self-preference bias, we used GPT-4o[9]. See Appendix H.2 for an example prompt.

In Table 8, we report the percentage of pairs kept after the filter in the Enhanced-Corrupted (C-O), Enhanced-Original (E-O), and Original-Corrupted (O-C) directions, as well as the number of training and dev samples after filtering. Intuitively, since the distance between

---

[6]2013-2021 data accessed at

[7]Verified by moderators with photo of self with credential documents. See `https://www.reddit.com/r/AskDocs/`.

[8]gpt-4o-2024-08-06

[9]gpt-4o-2024-08-06

enhanced and corrupted is the largest, the LLM-judge is the most likely to accurately detect the intended perturbation direction, whereas for the Enhanced-Original and Original-Corrupted pairs, more samples fail the LLM-judge filter. We can also see that among all the attributes, clarity has the lowest data quality before filtering.

| Attribute | E-C | E-O | O-C | # Train | # Dev |
|---|---|---|---|---|---|
| Accuracy | 99.3 | 98.3 | 70.9 | 11,994 | 1,155 |
| Answerable | 99.6 | 98.3 | 69.4 | 11,933 | 1,154 |
| DDX Bias | 99.8 | 86.9 | 94.5 | 12,548 | 1,223 |
| Clarity | 85.8 | 73.4 | 70.3 | 10,250 | 986 |
| Focus | 92.0 | 72.8 | 75.3 | 10,660 | 1,095 |
| Relevance | 99.2 | 73.0 | 91.2 | 11,756 | 1,142 |
| All | 96.0 | 83.8 | 78.6 | 69,141 | 6,755 |
| Coarse | 99.8 | 94.2 | 95.7 | 12,939 | 1,246 |

**Table 8:** Policy Fusion DPO models on attribute groups.

## C.2 Training Hyperparameters

We use Open-RLHF (Hu et al., 2024a) to train all models, and adopt the default hyperparameters. All models were trained on one A100 GPU and we list the hyperparameters of the final models below. Additionally, we experiment with different hyperparameter values (as listed in parentheses below) and find minimal differences in evaluation results.

Supervised fine-tuning:

- Epoch: 2
- Learning Rate: 5e-6 (1e-6, 1e-5, 5e-5)
- Warm up ratio: 0.03
- LR Schedule: `cosine_with_min_lr`
- Batch size: 256

DPO:

- Epoch: 1
- Learning Rate: 5e-7 (1e-6)
- Beta: 2 (0.1, 1, 4)
- Warm up ratio: 0.03
- LR Schedule: `cosine_with_min_lr`
- Batch size: 256

Reward modeling:

- Epoch: 1
- Learning Rate: 9e-6 (5e-7, 1e-6, 1e-5, 1e-4)
- Beta: 2
- Warm up ratio: 0.03
- LR Schedule: `cosine_with_min_lr`
- Batch size: 256 (4, 16, 64)

PPO:[10]

- Epoch: 1
- Learning Rate: 5e-7
- Warm up ratio: 0.03
- LR Schedule: `cosine_with_min_lr`
- Batch size: 256

---

[10]Note that while it's possible to use model parallelism to train 8B PPO models with qlora, we did not have the compute resources to train 8B PPO models with the same hyperparameters and settings as the 3B counterpart, so the experiments did not include comparisons with the 8B PPO models.

Training times on single A100 GPU with batch size 256:

- SFT: 4hr (3B model), 8.5hr (8B model)
- DPO: 10hr (3B model), 42hr (8B model)
- PPO: 48hr (3B model)

### C.3   MediQ Interactive Benchmark

We evaluate the downstream performance of the trained model by generating questions in a multi-turn clinical reasoning task using MediQ (Li et al., 2024). In MediQ, there is a patient agent and an expert agent interacting with each other, where the expert agent is provided some initial information in the beginning, and is expected to decide whether it wants to continue the interaction to acquire more information or terminate the interaction and provide a final answer. In this framework, the expert agent consists of three modules: abstention, question generation, and decision making. We fix the abstention and decision making modules using `meta-llama/Llama-3.1-8B-Instruct`, and replace the question generator with the model variants in our experiments. The goal of this evaluation is to show the effect of question quality on the final diagnostic accuracy. For reproducibility, we list the hyperparameters used in the MediQ interactive framework below:

- Abstention strategy: Scale
- Rationale generation: True
- Self-consistency: False
- Maximum interaction length: 15
- Temperature: 0.6

## D   Expert vs. Non-Expert Human-Written Questions

In `r/AskDocs`, members can upload credentials to acquire an expert flair (tag). In order to validate the quality of the LLM-judge, we aim to observe differences in the reported win-rates concerning questions generated by experts vs. non-experts. Since the original data is extremely scarce, there is limited samples where an expert and a non-expert respond to the same patient information, we design the following comparison scheme:

1. Starting with two sets of contexts, one with expert-written responses, one with non-expert written responses.
2. Using a variety of models—DPO-Coarse, ALFA-DPO-DataMix, ALFA-DPO-PolicyFusion, ALFA-PPO-DataMix, ALFA-PPO-PolicyFusion—to generate responses conditioned on each set of contexts.
3. Use our LLM-judge to compare the human written response to the model generated responses for each set of contexts.
4. Compute the win-rates of expert vs. model and non-expert vs. model.

Following the above procedure, we find that the win-rate of non-expert responses is **35.87%**, while the win-rate of expert responses is **50.52%**, suggesting that expert-written questions are of higher quality than the non-expert questions. Thus, this finding validates the relative accuracy of our LLM-judge. Additionally, this also shows that the responses generated by ALFA-aligned models are approaching human-expert quality.

## E   *MediQ-AskDocs* Task Construction

**Task Construction.**   To construct the *MediQ-AskDocs* task for clinical reasoning and use it in the MediQ interactive doctor-patient simulator, we need to parse each patient's information, collected from a Reddit conversation thread, into the following components: initial information, additional information, inquiry, options, correct answer.

Taking the entire conversation thread between the patient and the community member, including the initial post from the patient, as the patient's record, we prompt o1 to extract the initial information of the patient with few shot examples. Then, in a separate call, we

prompt o1 to extract the patient's inquiry—posts in r/AskDocs are often in the form that the patient posts a paragraph of their information and asking a health question—and the conclusion from the responder if any. Finally, we treat the parsed conclusion as the correct option, and prompt the model to generate three alternative wrong answers to the inquiry.

We generate multiple choice questions following the above approach for all 302 threads in the test split of the *MediQ-AskDocs* dataset to form the novel interactive healthcare QA task.

### E.1 Expert Annotations

**Task Setup.** We collected human expert annotations to validate the machine-generated multiple choice questions. To validate machine-generated questions, options, and the correct answer, the task included three questions: 1) plausibility of the generated question ("Yes" or "No"), 2) selecting a correct option out of four candidates with an additional option to select "None of the above", and 3) adding a free-text option if a plausible option is not listed. We randomly assigned maximum of 3 annotators per sample and paid 20 USD/hr. Participant recruitment process is detailed in Appendix F.

**Annotation Details.** Due to recruitment constraints, we collected three expert annotation per sample over 295 samples, excluding samples annotated during recruitment.

| Coefficient Name | Coeff | Interpretation | StdErr |
|---|---|---|---|
| Fleiss' Kappa | 0.58 | Moderate | 0.029 |
| AC1 | 0.65 | Substantial | 0.027 |

**Table 9:** Inter-rater agreement on multiple-choice question correct answers.

**Results.** On 298 samples, excluding 4 test questions, all questions were considered plausible and correct answers generated by o1 showed 85.9% accuracy based on majority vote on 298 samples. Upon examining agreement on 295 samples, excluding 7 samples used during recruitment, we see moderate to substantial agreement (Wongpakaran et al., 2013; Wong et al., 2021) as shown in Table 9.

## F Expert Manual Evaluation

Expert manual evaluation was conducted by a panel of three research team members with extensive medical training and qualifications: two MD-PhD specialists with expertise in radiology, pathology, gastroenterology, and oncology, and one MD resident specializing in pathology. The members were instructed to rank the proposed questions given context (patient post and previous conversation—questions and answers between medical expert and patient—if any) in an online medical consultation setting similar to r/AskDocs.

| *PO | POI | Win-rate |
|---|---|---|
| DPO | Data | 52 |
| | Policy | 41 |
| PPO | Data | **75** |
| | Reward | 74 |
| | Policy | 40 |
| Human | | 47 |

**Table 10:** Win-rate of each model variation over baseline model for different attribute integration strategies based on majority vote.

### F.1 Ranking Annotation Task Setup

We randomly selected 100 samples and asked our panel to rank 7 different questions per sample from best to worst follow-up questions. The seven question variations included one

| *PO | POI | AC1 | CI Lower | CI Upper | p-value | PA | PE |
|------|--------|-------|----------|----------|----------|-------|-------|
| DPO | Data | 0.129 | -0.090 | 0.349 | 0.245 | 0.520 | 0.449 |
| | Policy | 0.240 | 0.047 | 0.434 | 0.016* | 0.620 | 0.500 |
| PPO | Data | 0.721 | 0.594 | 0.848 | <0.001*** | 0.780 | 0.211 |
| | Reward | 0.680 | 0.543 | 0.817 | <0.001*** | 0.750 | 0.219 |
| | Policy | 0.224 | 0.028 | 0.419 | 0.025* | 0.610 | 0.498 |

*$p<0.05$, **$p<0.01$, ***$p<0.001$

**Table 11:** Gwet's AC1 inter-rater reliability results between Human and LLM Judge on win-rate over baseline by model variation. PA denotes percent agreement and PE denotes expected percent agreement by chance.

human-written question and six model generated questions under different experimental setup (Base Model, ALFA DPO Data, ALFA DPO Policy, ALFA PPO Data, ALFA PPO Policy, ALFA PPO Reward). The questions were presented in a randomly shuffled order, and annotation platform was constructed to not allow ties.

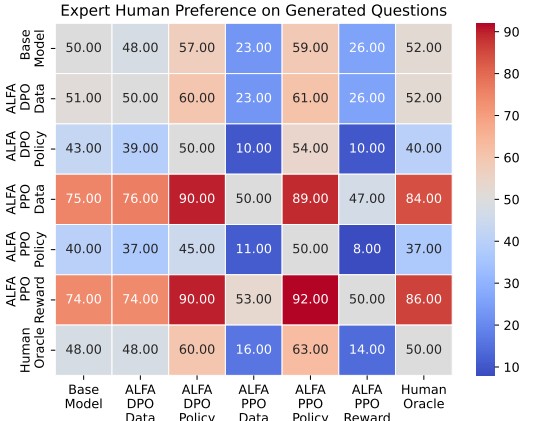

**Figure 7:** Expert preference ranking results showing pairwise win-rates. Models on y-axis are compared to the models on x-axis (e.g., ALFA PPO Data has 75% win-rate over Base Model).

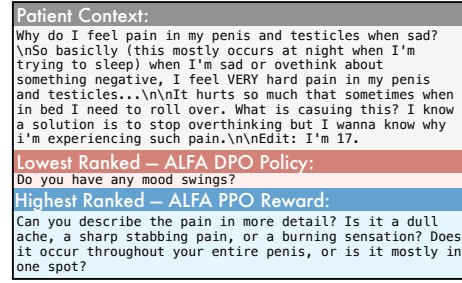

**Figure 8:** An example of majority annotated best question (PPO Reward) and worst question (DPO Policy).

## F.2 Results

Similar to the results discussed in §6.4, human annotation win-rate shows preference towards ALFA PPO Data and ALFA PPO Reward, trained on fine-grained reward fusion as shown in Table 10. Interestingly, while on automatic evaluation on medical diagnostic accuracy, ALFA DPO Policy outperformed other integration strategies, we observe that it only outperforms ALFA PPO Policy. This diverging outcome could indicate various factors, including stylistic preferences (Zhang et al., 2024b) that might influence annotation decisions which require further investigation. As noted in §6.4, our medical diagnostic accuracy evaluation relies on patient provided information; however, as shown in an example in Figure 8, the best quality question from ALFA PPO Reward is more open-ended and targets multiple aspects, which might not have been answered by the patient in our data.

**Inter-rater Agreement** Medical experts had a substantial pairwise ranking agreement of .481 Gwet's AC1 score. The experts also showed high agreement with LLM judge, especially for model variations with higher win-rate. Experts noted that questions with lower win-rate showed similar quality leading to random assignment of preferences as ties were not allowed on our platform, which could explain the lower agreement for model variations with lower win-rate.

| Context | |
|---|---|
| **Patient**: In Hospital Again...Need Real Life Dr. House 40m, white, non smoker casual drinker. I am currently in a hospital bed on a Heparin Drip. I have Vascular, Hematology, oncology, and even Gastro all stumped. Basically, I am under attack by blood clots but no one has a clue why. A year of this. Long story short - calf DVT 10/20. Found May Thurner, stent placed, clot resolved, moved on with life. Taken off thinners. July - spinal surgery. 3 weeks later, blood clots EVERYWHERE, spent a week in hospital had thrombectomy. Simple answer - surgery caused clots (even though ive had many surgeries before with no clots). Put on Xarelto. November - found more clots even on Xarelto. Do tPA lysis catheter puts me in hospital another week. Clears clogged stents, put on Lovenox. 2 weeks later - more clots. Just had another thrombectomy. I have tested negative for everything. Negative for all genetic factors, negative for rare clotting disorders. And yet, here I am. CT Scan of chest and pelvis no tumors. Lymph nodes swollen but šubcentimeterïn the area where all the clotting keeps happening. Everyone is stumped. They won't let me out of the hospital. I feel fine except for my leg which has been through alot. Only info to go off of - I have been anemic off and on, Monocyte % slighly high, folate very low, absolute immature gran count high. I am here in the hospital. I can ask anything. Give me your wildest ideas. Something to have them test. I want to go home but also desperate for a diagnosis. Thank you! PS: WOW- thank you all. Updates from what I've relayed off of this thread - hematology is now done with me unless something changes. They say they've tested for every clotting disorder and looked as much as they could for underlying cancers. They say definitively it is not APS. Update 2 (I don't think anyone is still checking this but just in case) - Tumor marker test was negative but they did find a precancerous polyp from the colonoscopy. **Doctor**: Yup, was wondering for their clotting disorders if they tested for connective tissue disease and things like antiphospholipid, lupus anticoagulant etc **Patient**: Yes. They said i had a weak positive for APS. They dont think its that but said they were going to run some more tests but havent yet | |
| ALFA-PPO-Data | Have you had any of the following symptoms in the past 2 weeks before the clots occurred: pain or swelling of the chest, abdomen, or groin, difficulty breathing, coughing or fever, pale gums, yellowing of the skin or eyes, nosebleeds or bleeding gums? |
| ALFA-PPO-Reward | Can you describe the anemia you've experienced? Was it constant, or did it only occur during periods of clotting, and if so, how often did it occur? Was it painful or associated with any other symptoms? |
| ALFA-PPO-Policy | Did you ever get the LVT? |
| ALFA-DPO-Policy | And have you had your thyroid checked? |
| Base | So they ran the APS test but aren't convinced it's the cause. In that case, here are some wild ideas to consider: 1. **Autoimmune response to something else**: Despite having low folate, maybe the issue isn't with folate deficiency but with an autoimmune response to something else. Could it |
| ALFA-DPO-Data | That's the problem - a weak positive is not conclusive, and they're still awaiting more results or further testing to rule it out or confirm it. Okay, wild ideas time! Here are a few potential test ideas, not that they're guaranteed to work, but might be considered: 1. **Cardiolip |
| Human | Did you or your sister have any negative tests before being diagnosed with lupus / APS? |

**Table 12:** Qualitative example of generated questions. The questions are ordered from best (ALFA-PPO-Data) to worst (Human) quality based on human annotation according to majority vote.

## G  Qualitative Analysis

Table 12 shows an example of a multi-turn conversation and generated follow-up questions ordered by ranking of best to worst quality as denoted by our team of medical experts. ALFA-PPO-Data ranked best out of all options and asks a comprehensive and targeted follow-up question. Moreover, the example shows that even while prompted to ask questions, and trained to do so, asking follow-up questions is a difficult task as shown by the response by base model and ALFA-DPO-Data.

## H  Prompts

### H.1  Counterfactual Perturbations

```
You are a medical assistant and your task is to rewrite medical questions
posted to an online health forum to vary some of their properties. The goal
is to generate these diverse counterfactual questions to study the properties
of clinical questions. You will be given a patient's post, and the original
clinician response, and you should rewrite the clinician response according
to the instructions below.
***PATIENT POST*** title post
***CLINICIAN RESPONSE*** question
***INSTRUCTION*** Rewrite the clinician response so that it is less clear/more
ambiguous for the patient, while keeping everything else constant.  The
definition of this property and what it means for this property at varying
scales are given below:
Definition: The ease with which a reader can understand the intent and meaning
of the question. A clear question avoids ambiguity and vagueness, providing
enough detail to prevent misunderstanding, while avoiding excessive complexity
or overloading with jargon. Very ambiguous: The question is highly ambiguous,
vague, or disorganized, making it very difficult to understand what the asker
is seeking. The question may lead to multiple interpretations and confusion.
Somewhat ambiguous:  The question is somewhat ambiguous or vague and may
include overly complex phrasing. It requires significant effort to interpret.
In-between:  The question is mostly understandable but could benefit from
rewording or simplification to remove partial ambiguity or excessive jargon.
Somewhat clear: The question is generally clear, with minimal ambiguity, and
can be understood by a layperson. There is little chance of misunderstanding.
Very clear: The question is entirely unambiguous, easy to understand, and
structured in a logical, concise manner. No jargon or unnecessary complexity.
Additional Tips for Clear Questions Use specific time frames:  Instead of
"lately," try "in the past week" or "since your last visit." Break down
complex questions: If a question could be answered in multiple ways, consider
asking two separate questions. Avoid medical jargon: Use plain language that
patients without a medical background can understand.
Please make the rewritten question more realistic - something that clinicians
would ask in an actual patient interaction.
Return the rewritten question ONLY and do not include any other text.
***REWRITTEN RESPONSE***
```

## H.2  Automatic Synthetic Data Verification

**SYSTEM**: Please act as an impartial judge and evaluate the quality of the responses provided by three medically trained AI assistants in a medical interaction. Carefully read the questions being asked by these expert systems as a response to the medical interaction and rank them in the provided dimensions. Begin your evaluation by comparing the three responses and provide a short explanation. Avoid any position biases and ensure that the order in which the responses were presented does not influence your decision.  Do not allow the length of the responses to influence your evaluation. Ignore possible spelling or grammar mistakes and focus only on the content of the text. Be as objective as possible. The only ranking choice is ">" (greater than). For each dimension listed, provide your answer in the following example JSON format: { "dimension_name": { "ranking": "A  B  C", "reasoning": "Provide a clear and concise explanation for your ranking decision here." } }
**USER**: Please carefully review the previous interaction below, which includes patient post, title, and subsequent responses if any. ***PREVIOUS MEDICAL INTERACTION*** prev_context ***MEDICAL AI QUESTIONS TO PATIENT*** - **Question A:** question_a - **Question B:** question_b - **Question C:** question_c ***EVALUATION DIMENSIONS*** dimensions ***SUPPLEMENTARY INFORMATION*** To help you evaluate the questions, please refer to the provided additional information regarding the final conclusion of this patient's case below: Final diagnosis: final_diagnosis Conclusion: conclusion

### H.3  *MediQ-AskDocs* Task Construction

**SYSTEM**: You are a experienced expert working in the field of medicine education. Based on your understanding of basic and clinical science, medical knowledge, and mechanisms underlying health, disease, patient care, and modes of therapy, you are given a patient case and you are tasked to parse the patient's inquiry into a multiple choice question. The generated multiple choice should consist of a question and 4 options, which could be answered by the given patient conversation. Base your response on the current and standard practices referenced in medical guidelines. The created question should be answerable only with the patient information, rather than testing some hardcore scientific foundational knowledge recall. The questions should be faithful to the original patient's inqiury in their post. The correct answer should be correct, and the distractors should be plausible. The correct answer should be evenly distributed among the available options to enhance the quality and reliability of the questions. The output should be in json format.

**USER**: You could use some parsed auxiliary information such as the final diagnosis and conclusion. Make sure that the multiple choice question you generate is not too easy but also not impossible to answer. Based on this patient record, faithfully generate a multiple choice questions according to the patient inquiry and store them in the following json format:

```
{
"question": [generated question 1],
"optionA": [option A],
"optionB": [option B],
"optionC": [option C],
"optionD": [option D],
"correct_answer": [A or B or C or D]
}
```

After you generate the question, do a round of revision. In your revision, you should:
1. Identify any medical inaccuracies in your first response, corrsect them if any exists.
2. Make sure the question is what the patient is asking for or concerned about in their post.
3. The correct answer is indeed correct, if none of the options are correct or more than one options are correct, revise the options to improve the question.
4. Ensure that the correct answer is in a random position among the available options (shuffle if necessary) to enhance the quality and reliability of the questions.
5. Guarantee that the json output is parsable.

Respond with the final revised question in the json format and NOTHING ELSE.

