# OpenReview forum: "ALFA: Aligning LLMs to Ask Good Questions A Case Study in Clinical Reasoning"
_colmweb.org/COLM/2025/Conference — COLM 2025_

### Official Review · Reviewer_3HyA · 2025-05-12

**Rating:** 8
**Confidence:** 3
**Ethics Flag:** 1

**Summary:**

This work introduces a framework for LLMs to ask meaningful questions by tuning the model with the preferences on which question is better than the other question.
Specifically, the authors define 6 attributes, which are clarity, focus and answerability for general questions, as well as accuracy, relevance and DDX bias for clinical domain questions.
Based on these attributes, the authors prompt an LLM to generate pairs of enhanced (better) questions and corrupted (worse) questions regarding each individual attribute, and suggest to tune the model using these preference pairs with standard preference optimization algorithms such as DPO and PPO.

**Questions To Authors:**

None.

**Reasons To Accept:**

1. The paper is well-written.
2. The work is well-motivated. Especially, since the public datasets are not largely available in healthcare domains due to the privacy issues, researchers to develop question answering systems usually rely on LLM-generated QA pairs. In such scenario, the LLMs tuned by the suggested framework will serve as a useful tool to generate clinically meaningful dialogues, which can help improve quality of the resulting QA datasets by grounding the content in medically appropriate knowledge and style.
3. The results show that the model aligned with the generated preferences pairs outperforms baseline models (llama-3.2-3b-Instruct and llama-3.1-8b-Instruct), demonstrating the need of explicit framework for LLMs to produce good questions.
4. The ablation studies are well-structured.

**Reasons To Reject:**

1. The dataset is collected from an open internet ($\texttt{r/AskDocs}$), which seems not convincing with regards to its conversation quality.

---

> ### Author Response · Authors · 2025-06-03
>
> We would like to thank the reviewer for thoughtful comments. We are especially grateful for your positive comments about the motivation of our work, ALFA’s outstanding performance, and the rigor of our ablation studies.
>
> We agree that good quality data is important. We want to highlight three considerations we employ to ensure the quality of the real data and, in turn, our real+synthetic dataset.
>
> 1. The subreddit has strict policies in place against misinformation or harmful comments and a verification process for medical professionals. Many users are marked as “experts” and both human and bot moderators are actively supervising the quality of the medical feedback provided by the users in the community.
>
> 2. The comments, randomly chosen from the subreddit, have been found to be on average acceptable by licensed healthcare professionals in a previous study [1]. We have also conducted in-depth consultations with collaborators from the medical school to ensure that the conversations are of acceptable quality and reasonable realism.
>
> 3. We parsed and included samples with positive feedback from the original poster and presence of conclusion to ensure conversation quality as described in the Appendix.
>
> We will clarify this better in the paper to enhance the believability of our results and we hope that the above points address the concern regarding dataset quality. We sincerely thank the reviewer for their thoughtful and positive feedback and welcome any further discussions or questions.
>
> [1] Ayers, J. W., Poliak, A., Dredze, M., Leas, E. C., Zhu, Z., Kelley, J. B., ... & Smith, D. M. (2023). Comparing physician and artificial intelligence chatbot responses to patient questions posted to a public social media forum. JAMA internal medicine, 183(6), 589-596.

---

> > ### Comment · Reviewer_3HyA · 2025-06-09
> >
> > Thank you for addressing my comments.

---

### Official Review · Reviewer_cdw7 · 2025-05-13

**Rating:** 6
**Confidence:** 5
**Ethics Flag:** 1

**Summary:**

The paper proposes ALFA, which trains models that generate clinical questions with measurably better alignment on multiple assessment criteria. The work does this by breaking down the assessment of a question into multiple important measurable characteristics grounded in theory. Then generated a dataset by agumenting the existing dataset to get a preference dataset on all the selected attributes. The authors then show how to use this dataset to instruction-tunned models.

**Reasons To Accept:**

- The authors use a theory grounded approach to creating assessment dimensions of clinical questions.
- Strong performance gains when using ALFA
- Synthetic Counterfactual Data Generation with Controlled Perturbations is an interesting and scalable approach to preference dataset creation.
- Ablation studies to investigate data synthesis direction, attribute inclusion, integration strategy (data/reward/policy fusion), and model size
- Modular design (decompose → synthesize → align) provides a replicable recipe for future work in other expert domains
- Authors contribute a high-quality dataset combining real-world with synthetic preference pairs and expert annotations

**Reasons To Reject:**

Major Concerns:
- No direct comparison with LLMs
    - The paper compares ALFA primarily against instruction-tuned LLaMA-based models and SFT baselines. It does not benchmark against existing medically-tuned models (e.g., MedPaLM, PMC-LLaMA, or BioMistral) or other general purpose strong models like GPT4 or Gemini, which could provide a stronger empirical context for the results.
    - There is one line in the paper mentioned without any details that ALFA performs better than GPT4o in downstream utility. This statement is not useful for the readers without any details.
- Potential Page Limit Violation because the Table 7 is present in the appendix, but referenced in the results.

Minor concerns:
- Lack of Real Clinical Dialogues in the evaluation or in the dataset
   - The paper uses Reddit QA in the dataset, and the interactive clinical tool (using LLM-based agents to roleplay), but the real-world setting is not explored in the work.
- ALFA requires manual, domain-expert-defined attributes, which may not scale easily to other domains or more complex tasks. The paper acknowledges this.

---

> ### Author Response · Authors · 2025-06-03
>
> We thank the reviewer for their thoughtful comments. As noted in the review, our framework is designed to integrate fine-grained, domain specific attributes, which show strong improvement, and the modular design allows easy adaptation to other domains in future works.
>
> > **Direct comparison with LLMs**
>
> Thank you for raising this concern. We do actually compare with one general purpose LLM and find that we outperform GPT-4o (**L288**) on the interactive diagnostic task of MediQ-AskDocs. GPT-4o has an accuracy of 79.8% on a randomly selected 9.4% lower than the ALFA-DPO 8B model. We will update the potential confusing wording of “downstream task utility” to clarify this confusion. We have added results for Gemini-2 as well per your suggestion, and obtain an accuracy of 79.8%.
>
> And as we discussed in the related works, local medical models such as PMC-LLaMA and BioMistral, and Meditron, are known to be optimized for medical knowledge, rather than diagnostic (conversational) capabilities and are often lacking in instruction-following ability. Therefore, it is difficult to correctly prompt them to provide high quality questions, we thought that it would not be a fair comparison. But we can include the performance of these models (reported below) in the updated draft. Note that Med-PALM is not accessible for research purposes.
>
> |Model|Diagnostic Accuracy|
> |--|--|
> |DPO-ALFA|88.1|
> |GPT-4o|79.8|
> |Gemini-2|79.8|
> |o3-mini-low|71.2|
> |Alpacare-7B|74.8|
> |Alpacare-13B|73.5|
> |ClinicalCamel-70B|70.5|
> |Meditron-7B|70.2|
> |MedAlpaca-13B|68.5|
>
> > **Referring to Table in Appendix**
>
> We thank the reviewer for catching this slight typo. We actually meant to refer to Table 6, which is the performance of the models when removing certain attributes (presented on Page 8). We will fix the typo in the updated draft.
>
> > **Clarification on the use of real-world data**
>
> We use Reddit QA data—a real-world dataset of medical questions and human interactions—due to the lack of publicly available clinician-patient dialogue datasets focused on question-asking. While real-world evaluations, such as clinical trials, are essential for works that focus on medical communication, they require a deeper collaboration with medical researchers as well as patients, which is beyond the scope of the current project. As we also pointed out in the Ethics Statement section, ALFA is intended as a technical contribution to the field of computer science rather than a standalone clinical tool.
>
> > **Manual selection of domain-specific, expert attributes**
>
> We first want to highlight that the ALFA pipeline itself is attribute-agnostic—different attributes can even be tested for their utility in the desired task as we did in Table 6 of the Ablations subsection (S 6.4). Regarding attribute selection, the general question asking attributes (e.g., focus, clarity, answerability) can be generalized to other domains. We leave attributes that require domain specific expertise for future exploration and collaboration with respective experts (e.g., legal, teaching, mental health). For future work, there are also opportunities to explore automatic attribute selection (potentially starting with lower stake domains) using LLMs.
>
> We hope that our response addresses the reviewer’s concerns and any potential confusions. We welcome any additional discussion or questions. If we have addressed any of the questions or concerns you have about this work, we would greatly appreciate it if the reviewer could consider raising their scores to reflect the discussion.

---

> ### Author Response · Authors · 2025-06-07
>
> Dear reviewer,
>
> Thank you so much again for your original review.
>
> Additionally, we have added results for an even more comprehensive list of medical models, including the PMC_LLaMA model requested in the original review. We hope that these results better contextualize ALFA's superior performance and strengthen our paper.
>
> - Meditron-70B: 72.5
>
> - PMC-LLaMA-7B: 71.9
>
> - PMC-LLaMA-13B: 69.9
>
> We believe that we have address all questions and clarification requests in the original review. Please let us know if you have any additional questions or concerns. The discussion period ends soon: we would really appreciate it if you could let us know your thoughts regarding our answers to your comments and questions, and consider raising your scores if we have cleared any of the confusions.
>
> Thank you so much.

---

> ### Author Response · Authors · 2025-06-10
>
> Dear Reviewer cdw7,
>
> As the rebuttal period is coming to an end, we were wondering if our new experiments and results addressed your questions? If so, we would really appreciate it if you would consider updating your scores to reflect the helpful discussion.
>
> Best,
> Authors

---

### Official Review · Reviewer_yWPb · 2025-05-13

**Rating:** 8
**Confidence:** 4
**Ethics Flag:** 1

**Summary:**

This paper presents an approach for better question generation in the context of clinical reasoning and LLM-assisted diagnosis. There are three key concepts in the proposed approach:

(1) Provide a better, more clear definition of what better quality questions should be, both in general and in the context of clinical reasoning, based on a set of grounded attributes extracted from related work. Examples of these are clarity and focus (for the general context) and diagnostic for the medical context.

(2) With these attributes in mind, utilize an LLM to synthesize questions by modifying existing seed questions making them more or less satisfactory for a given "good question" fine-grained attribute, e.g. making a question more or less "clear". This results on "enhanced" and "corrupted" versions of each seed question

(3) Utilize these pairs of synthesized questions as preference data for aligning LLMs, and test several preference-based optimization techniques to train models to ask better question along these fine-grained attributes.

To test this, the paper proposes to leverage the `r/AskDocs` reddit forum, which contains interactions verified by medical experts. The empirical framework proposed on this data is sound and complete, with analysis of the impact of several key decisions in the including but not limited to:

- Filtering applied to the data generated as a result of step (3), performed via using an LLM-as-a-judge
- Comparison of different LLM alignment techniques, including SFT and preference-based training. Critically, these are applied to exactly the same data.
- Different decomposition of "good question" attributes from fine-grained (clarity, medical accuracy, etc.) to  "coarse" (just good or bad)

The resulting models are tested both on in-domain data as well on out-domain data, the latter being the result of a held-out portion constructed using the source data. Several key insights are derived from the empirical results, indicating which data filtering and training techniques lead to more aligned and/or better quality output.

**Questions To Authors:**

- In Table 1, both 3B and 8B base models exhibit a win-rate performance of exactly 50%. I find this a bit suspicious, could you confirm that these are empirical values, and if so, how these numbers were obtained?

- In line 245: where does the number 56.62% come from?

- I wonder if it would be possible to have a small study on the impact of the LLM used for data curation, specially since the paper relies on GPT-4o a lot. At this point, do we require a cutting-edge model for these steps, or can we get by with a cheaper, perhaps local model?

- Finally, it would be helpful if the paper could provide estimates on the amount of compute and budget used for the experiments, to help inform future work on the trade-offs in this sense.

**Reasons To Accept:**

- Overall, I think this paper is a solid contribution to COLM. It tackles a relevant research problem and proposes a sound and effective solution.

- The insights derived from the experiments presented allow us to learn which parts of the proposed pipeline are critical and which training schemes lead to better output. I learned a significant amount from this paper, and the lessons learned here will sure inform future work, so I think this paper can have a significant impact in our community.

- The paper is generally clear and well written.

**Reasons To Reject:**

None, but please see my questions below.

---

> ### Author Response · Authors · 2025-06-03
>
> We thank the reviewer for insightful comments. We are excited that you found the **paper’s topic relevant** with a **sound and effective solution** to the **proposed problem with a significant impact**, and found no reasons to reject. We hope to answer the reviewer’s questions below.
>
> > **In Table 1, how were win-rate performance of exactly 50% obtained?**
>
> The values in Table 1 for 3B and 8B are not empirical values. Since win-rates are compared against the base model, the values filled for the base model are simply indicating that when provided with two equal quality samples, the win-rate would be 50. We will clarify this in the updated draft.
>
> > **In line 245: where does the number 56.62% come from?**
>
> For the 8B model, the base performance is 72.52, the best (ALFA-DPO) performance is 88.08, so the diagnostic error rate is 100-72.52=27.48 and 11.92, respectively, resulting in a error reduction rate of 1-11.92/27.48 = 0.5662. We report this number because diagnostic error rate is a metric commonly used when discussing clinical support tools, rather than diagnostic success rate as commonly used in CS/NLP literature. We can include the gain on diagnostic success rate (88.08/72.52-1=0.215) in the updated draft as well for clarity.
>
> > **Impact of LLM used for data curation and exploration of smaller models**
>
> We highly agree with the reviewer’s suggestion to investigate using smaller models to serve parts of (or the entire) pipeline. For example, a reasonable starting point could be using a weaker model to generate the corrupted direction of the paired data (e.g. less clear, less focused, etc.), then apply some post-filtering verification strategies to ensure quality.
>
> In our experiments, we use GPT-4o for most steps but also recognize the limitation of high cost, so Llama-3.1-405B-FP8 was used for massive synthetic paired data generation (L361). Although it’s still a large model, it is possible to be deployed locally with a reasonable amount of compute. We further use GPT-4o to validate its generation. For other components of the pipeline and evaluation, we use GPT-4o because it had high performance in medical tasks [1] and had been used in various medical studies [2,3].
>
> > **Amount of compute and budget used for the experiments**
>
> Each training run is done on a single a100 GPU with a batch size of 256 (micro batch size of 1) for 1 epoch. The compute used for the experiments were:
>
> - SFT: 4hr for the 3B model, 8.5hr for the 8B model.
>
> - DPO: 10hr for the 3B model, 42hr for the 8B model.
>
> - PPO: 48hr for the 3B model.
>
> We will include this information and detailed hyperparameters in the updated draft.
>
> [1] Nori, H., King, N., McKinney, S. M., Carignan, D., & Horvitz, E. (2023). Capabilities of gpt-4 on medical challenge problems. arXiv preprint arXiv:2303.13375.
>
> [2] Jo, E., Song, S., Kim, J. H., Lim, S., Kim, J. H., Cha, J. J., ... & Joo, H. J. (2024). Assessing GPT-4’s performance in delivering medical advice: comparative analysis with human experts. JMIR Medical Education, 10(1), e51282.
>
> [3] Goh, E., Gallo, R. J., Strong, E., Weng, Y., Kerman, H., Freed, J. A., ... & Rodman, A. (2025). GPT-4 assistance for improvement of physician performance on patient care tasks: a randomized controlled trial. Nature Medicine, 1-6.

---

> > ### Comment · Reviewer_yWPb · 2025-06-10
> > **Reply to authors**
> >
> > Thank you for taking the time to address the concerns I raised in my review. After reading the reply, my overall impression and assessment of this paper did not change significantly.
> > Hence, I am keeping my original score.

---

### Official Review · Reviewer_aB4k · 2025-05-15

**Rating:** 7
**Confidence:** 3
**Ethics Flag:** 1

**Summary:**

The main contribution of this paper is the ALFA framework, which aims to improve the question-asking ability of large language models in expert domains, focusing on clinical reasoning.

The paper also contributes the MediQ-AskDocs dataset, which includes 17k real-world clinical dialogues and 80k synthetic attribute-specific preference pairs.

The paper presents a comprehensive analysis and comparisons among integration strategies (DPO, PPO, reward vs. policy fusion).

**Questions To Authors:**

How does ALFA handle conflicting signals between attributes (e.g., clarity vs. medical accuracy)? Could there be dynamic weighting mechanisms in place?

**Reasons To Accept:**

It presents rigorous empirical validation.

Generalizable framework applicable beyond clinical domains.

**Reasons To Reject:**

Reliance on synthetic data and LLM judgments could introduce bias or hallucinations.

Attribute selection is manual and might not generalize without domain-specific expertise.

---

> ### Author Response · Authors · 2025-06-03
>
> We thank the reviewer for the positive review and thoughtful questions, especially focusing on **our rigorous empirical validation** and **generalizable framework** and the **recognition of our dataset** contribution.
>
> > **Verifying quality of synthetic data and LLM judgments**
>
> We understand the concern about synthetic data. To ensure the data quality, we verified using an automatic measure using GPT-4o as a judge and filtered the dataset based on the verification results. Furthermore, we generated the perturbed question pairs conditioned on the original human-written one to limit the risk for hallucination and bias. While there is still a risk for hallucination and bias, which should be investigated further in its effect in training, it was out of scope for the current project to dive deeper into this topic.
>
> > **Manual attribute selection and generalizability**
>
> We agree with the reviewer’s inquiry about attribute selection and we want to highlight that the ALFA pipeline itself is attribute-agnostic and can be adapted to other domains—different attributes can even be tested for their utility in the desired task as we did in **Table 6** of the Ablations subsection (**S 6.4**). Furthermore, general attributes such as focus, clarity, and answerability are broadly applicable and can often be transferred across domains. However, for domain-specific applications—such as legal, education, or mental health—attribute selection should be guided through close collaboration with domain experts to ensure both safety and usability. In such contexts, safety may take precedence over generalizability. We therefore view this as an important direction for future work, including the potential for automated attribute selection in lower-stakes settings.
>
> > **Dynamic weighing of attributes**
>
> We also think that dynamic weighing of attributes for resolving conflicting signals is a promising direction that was out of scope for the current project. However, our current framework makes it possible to integrate such a meta learning mechanism, which we believe is another strength of ALFA.
>
> Finally, we sincerely thank the reviewer for their feedback and appreciate any followup discussion if there are any remaining questions or concerns. We will include the points above in the discussion section of the updated draft. If we have addressed any of your concerns, we would appreciate it if the reviewer could consider raising their rating.

---

> > ### Comment · Reviewer_aB4k · 2025-06-09
> >
> > The authors have addressed all my concerns.

---

### Comment · Area_Chair_kGhX · 2025-06-05
**Official Comment by Area Chair kGhX**

Dear Reviewers,

As we near the end of the discussion period, I noticed there hasn’t been any discussion on this paper. If you haven’t already, please review the other reports and the author’s rebuttal.

Your input is essential to help us reach a decision.

Thank you,
AC

---

### Decision · Program_Chairs · 2025-07-08

**Decision:**

Accept

**Comment:**

This paper proposes ALFA, a modular framework for improving question generation in clinical contexts using attribute-specific preference data and alignment techniques. Reviewers agree that the work is well-motivated, clearly written, and presents strong empirical results with rigorous ablations. The use of grounded attributes and synthetic counterfactual data is seen as a scalable and insightful approach, and the dataset contribution is appreciated. However, some reviewers raise concerns about the reliance on synthetic and Reddit-based data, and the lack of comparison with state-of-the-art models like MedPaLM or GPT-4. There are also questions about the scalability of manually defined attributes and the absence of real clinical dialogues in evaluation. Despite these issues, the paper is generally viewed as impactful, and reviewers believe it will inform future research in expert-domain LLM alignment. I recommend acceptance.